# The Influence of Myeloid-Derived Suppressor Cell Expansion in Neuroinflammation and Neurodegenerative Diseases

**DOI:** 10.3390/cells13070643

**Published:** 2024-04-06

**Authors:** Lorenza Tamberi, Alessia Belloni, Armanda Pugnaloni, Maria Rita Rippo, Fabiola Olivieri, Antonio Domenico Procopio, Giuseppe Bronte

**Affiliations:** 1Department of Clinical and Molecular Sciences (DISCLIMO), Polytechnic University of Marche, 60121 Ancona, Italy; l.tamberi@pm.univpm.it (L.T.); armanda.pugnaloni@univpm.it (A.P.); m.r.rippo@univpm.it (M.R.R.); f.olivieri@univpm.it (F.O.); a.d.procopio@univpm.it (A.D.P.); g.bronte@staff.univpm.it (G.B.); 2Clinic of Laboratory and Precision Medicine, National Institute of Health and Sciences on Ageing (IRCCS INRCA), 60124 Ancona, Italy

**Keywords:** myeloid-derived suppressor cells (MDSCs), neuroinflammation, neuro-immune disease, neurodegenerative disease, Alzheimer’s disease, Parkinson’s disease, amyotrophic lateral sclerosis, multiple sclerosis

## Abstract

The neuro-immune axis has a crucial function both during physiological and pathological conditions. Among the immune cells, myeloid-derived suppressor cells (MDSCs) exert a pivotal role in regulating the immune response in many pathological conditions, influencing neuroinflammation and neurodegenerative disease progression. In chronic neuroinflammation, MDSCs could lead to exacerbation of the inflammatory state and eventually participate in the impairment of cognitive functions. To have a complete overview of the role of MDSCs in neurodegenerative diseases, research on PubMed for articles using a combination of terms made with Boolean operators was performed. According to the search strategy, 80 papers were retrieved. Among these, 44 papers met the eligibility criteria. The two subtypes of MDSCs, monocytic and polymorphonuclear MDSCs, behave differently in these diseases. The initial MDSC proliferation is fundamental for attenuating inflammation in Alzheimer’s disease (AD), Parkinson’s disease (PD), and multiple sclerosis (MS), but not in amyotrophic lateral sclerosis (ALS), where MDSC expansion leads to exacerbation of the disease. Moreover, the accumulation of MDSC subtypes in distinct organs changes during the disease. The proliferation of MDSC subtypes occurs at different disease stages and can influence the progression of each neurodegenerative disorder differently.

## 1. Introduction

Myeloid-derived suppressor cells (MDSCs) are a heterogeneous innate immune cell population known for their potent immunomodulatory activity. MDSCs are implicated in regulating immune responses in many diseases, including cancer, autoimmunity, and chronic inflammation. Although the role of these cells has been extensively studied in cancer, in recent years the involvement of MDSCs in chronic inflammatory conditions has also been highlighted, so MDSCs have become an attractive research area. Nevertheless, unlike cancer, limited and sometimes conflicting data are available about the role and function of these cells in neurodegenerative diseases, where they appear to behave as a double-edged sword. Even though the central nervous system (CNS) has been considered immune privileged for a long time, nowadays it is accepted that the immune and nervous systems are continuously subject to cross-talk [1]. The neuro-immune axis plays a crucial role both during physiological and pathological conditions. With the implication of MDSCs in chronic neuroinflammation, they are also involved in neurodegenerative disorders. MDSCs exert their immune-regulatory role in suppressing inflammatory responses by inducing T cell apoptosis in tumours and inflamed tissues. Indeed, MDSC expansion is well known in many inflammatory pathological conditions, and it should be considered a normal immune response to counteract chronic immune activation. On the other hand, uncontrolled MDSCs can worsen the status, indicating that they could be involved in the pathogenesis of certain conditions [2]. Therefore, their exact role remains controversial.

### 1.1. Phenotype of MDSCs

MDSCs express common myeloid markers and lack the expression of lymphoid markers. Specifically, MDSCs express different peculiar markers in mice or humans. In mice, MDSCs express Gr-1 and CD11b, and they are currently divided into two groups: (i) a monocyte morphology (M-MDSCs) exhibiting CD11b^+^Ly6C^hig^hLy6G^−^ phenotype; (ii) a granulocyte morphology (G-MDSCs) with a CD11b^+^ Ly6C^low^ Ly6G^+^ phenotype [3]. Whereas in humans, MDSCs express CD33 and CD11b, and lack expression of maturation markers, such as HLA-DR. In humans, MDCSs are subdivided into three populations. Two main subtypes are identified according to their morphology and the expression of monocytic (M-MDSCs) and granulocytic (G-MDSCs) or polymorphonuclear (PMN-MDSCs) markers: (i) M-MDSCs exhibit HLA-DR^–low^CD11b^+^CD33^+^CD14^+^CD15^−^ phenotype; (ii) PMN-MDSCs have a HLA-DR^−^ CD11b^+^CD33^mid^CD15^+^CD14^−^ phenotype [3,4]. The third subset of MDSCs, called early-stage MDSCs (e-MDSCs), displays CD33^+^CD11b^+^Lin^−^HLA-DR^−^CD14^−^CD15^−^ phenotype [4].

According to the classification, PMN-MDSCs and M-MDSCs differ in their phenotype. This dissimilarity results in their peculiar, although to some extent overlapping, functionality, which reflects their different roles observed under pathological conditions.

### 1.2. The Dual Role of MDSCs

MDSC expansion and the consequent immune suppression are important in physiological processes such as pregnancy [4], but they seem to be ineffectual in diseases. 

In inflammatory processes, the synthesis of pro-inflammatory factors, such as tumour necrosis factor alpha (TNF-α), interferon-gamma (IFN-γ), interleukin 6 (IL-6), and interleukin 1β (IL-1β), is commonly increased, and it is linked to the differentiation of suppressor cells. The two types of MDSC differentiate during two distinct but connected events. The presence of pro-inflammatory cytokines induces the differentiation of PMN-MDSCs. PMN-MDSC expansion modifies the microenvironments, which leads to the differentiation of M-MDSCs. MDSC activation depends on STAT3 activation, which has been demonstrated to occur in the presence of TNF-α, IFN-γ, and IL-6 pro-inflammatory mediators [5,6]. The peculiar expansion of the PMN-MDSC subset is induced by IL-1β, a member of the interleukin 1 family of cytokines [7,8], which is produced by activated macrophages, monocytes, and dendritic cells, among other immune cells. In inflammatory processes, the number of PMN-MDSCs seems to increase along with the disease severity [9]. 

Once differentiated, PMN-MDSCs execute double action; on one hand, they induce T regulatory cell (Tregs) differentiation and expansion, and, on the other, they are involved in the immunosuppression of T effector cells (Teffs). PMN-MDSCs express different factors, including programmed death-ligand1 (PD-L1), transforming growth factor β (TGF-β), interleukin 10 (IL-10), arginase-1 (ARG1), and reactive oxygen species (ROS) [10,11,12,13]. Through the activation of the programmed death-1 (PD-1)/PD-L1 pathway, PMN-MDSCs induce the differentiation and the expansion of Tregs, in the presence of TGF-β and IL-10 [10]. 

The immunosuppressive capability of both subtypes of MDSCs involves multiple mechanisms, including depletion of L-arginine, a crucial nutritional factor needed for T-cell proliferation, via ARG1 and increased production of ROS by the NADPH oxidase. The suppression seems to be exerted on Teffs, and not on Tregs [5].

Indeed, Tregs are expanded in chronic infection with respect to acute infection. In chronic infection, Tregs express CD103, a hallmark of the active state, and PD-1 receptor, through which they interact directly with PMN-MDSCs (PD-L1/PD-1 pathway). This interaction is crucial for the inhibition of apoptosis in Tregs, explaining the way through which PMN-MDSCs induce Treg differentiation and expansion. Activated Tregs exert double action; on one hand, they perform immunosuppressive activity on T helper 17 cells (Th17), via the PD-1/PD-L1 pathway, and indirectly via the expression of IL-10, an anti-inflammatory cytokine [14]. On the other hand, Tregs mediate the induction of M-MDSC differentiation, via the release of TGF-β [11]. 

In turn, it has been demonstrated that the expression of TGF-β bR2 (receptor type II) is enhanced in M-MDSCs in the presence of TGF-β. Moreover, M-MDSCs express ARG1 and nitric oxide (NO) through inducible NO synthase. M-MDSCs, along with PMN-MDSCs, also exert immunosuppressive activity on Teffs. M-MDSCs also produce IL-10, TFG-β, and IL-6 [11]. IL-10 could sustain the expansion of Tregs, thus acting as positive feedback. IL-6 is a pro-inflammatory cytokine which potentiates inflammation. Moreover, TGF-β and IL-6 are critical for the co-induction of pro-inflammatory Th17 cell differentiation from CD4+ T cells, although even IL-1β seems to be involved [15,16,17]. Emerging evidence indicates that Th17 cells and interleukin 17 (IL-17) are associated with chronic inflammation and pathogenesis of human neurodegenerative diseases. IL-17, being a potent pro-inflammatory cytokine, acts as a central regulator of inflammatory responses within the brain. Therefore, the excessive expansion and the prolonged accumulation of MDSCs may have a detrimental effect by enhancing the expansion of Th17 cells and IL-17 release, which can further lead to increased inflammation and worsen tissue damage in neurodegenerative diseases [17] (Figure 1).

Although IL-17 has been found to play a pivotal role in the pathogenesis of numerous inflammatory diseases in the CNS, little is known about its role in neurodegeneration and/or neurogenesis. IL-17 acts through the specific IL-17 receptor (IL-17R), which is widely expressed and binds IL-17 with high affinity. IL-17/IL-17R pathway activation results in phosphorylation of p38 MAPK and up-regulation of IL-6, interleukin 1 (IL-1), and NF- kB [18].

The expression of IL-17R has been detected within the CNS and upregulated under inflammatory conditions [19]. Particularly, IL-17R is expressed by neurons in the hippocampus, spinal cord, and neural stem cells, as well as in astrocytes and microglia [20,21]. In turn, IL-17 is expressed by infiltrated immune cells as well as glial cells in the hippocampus, motor cortex, and thalamus, as demonstrated in experimental autoimmune encephalomyelitis (EAE) animals [20].

In the hippocampus, IL-17 influences synaptic plasticity through the activation of its specific intermembrane receptor IL-17R in the CA1 region [20]. Moreover, in EAE animal models, IL-17 expression correlates with long-term potentiation (LTP) disruption in the acute phase of the pathology, suggesting its role in the modulation of neuronal transmission and thus in cognitive processes. Indeed, synaptic LTP represents a plastic phenomenon since it recapitulates the hallmarks of a biological learning process, and it is at the basis of memory engram formation, and, therefore, it is fundamental in cognitive processes. Cognitive impairment is a disabling concomitant of neurodegenerative diseases, including AD, PD, ALS, and MS. Considering that, MDSC expansion and the consequent IL-17 uncontrolled expression in nervous tissue, could have a role in the establishment of cognitive impairments. 

## 2. Materials and Methods

### 2.1. Source and Search Strategy

The search for articles was performed exclusively on PubMed entering the terms and combining them with Boolean operators (AND, OR) as follows: (“myeloid derived suppressor cell” OR “myeloid derived suppressor cells” OR MDSC) AND (“neurodegenerative disease” OR “Alzheimer’s disease” OR “Parkinson’s disease” OR “prion disease” OR “Amyotrophic lateral sclerosis” OR “multiple sclerosis” OR “motor neuron disease” OR “Huntington’s disease” OR “spinal muscular atrophy” OR “spinocerebellar ataxia”).

### 2.2. Data Collection and Sorting

Publication dates were not limited; all the results obtained by the research have been collected and classified according to the year of publication, journal, article type, experimental models used, and neurodegenerative disease debated. Once assorted according to these parameters, the results have been selected or excluded for analysis according to the eligibility criteria.

### 2.3. Study Selection Process

According to the aim of this review, we set out to consider only original articles concerning the changes of at least one MDSC subset in, at least, one neurodegenerative disease. All the results that did not meet these eligibility criteria were excluded.

Our search strategy produced 80 results, published between 2009 and 2023. Among these retrieved papers, we found 54 original articles, 19 reviews, 2 brief reports, 1 case report, 1 clinical trial, 1 editorial commentary, and 2 not-accessible papers. We excluded those that were not original articles and/or not accessible. Among the 54 remaining results, 10 did not meet the inclusion criteria, hence 44 articles concerning the MDSC changes in neurodegenerative diseases have been considered (Figure 2).

The analysis of all selected results highlighted the investigation of MDSCs in four different neurodegenerative diseases, namely AD (13.33%), PD (11.11%), ALS (2.22%), and MS (73.33%) (Figure 3A). No articles regarded MDSCs in motor neuron disease, Huntington’s disease, prion disease, and spinocerebellar ataxia. Some works used more than one experimental model (11.36%), while most of the studies were conducted using a single one (88.63%). The murine model approach is the most used (62.6%), followed by human specimens, like blood samples or *post-mortem* nervous tissues (28.83%), an in vitro approach using cells collected from patients or animal models (10.42%), and in silico using bioinformatic analysis (6.25%) (Figure 3B).

## 3. Results

### 3.1. The Immune System Participation in Neurodegenerative Diseases

In neuroinflammatory processes, immune and glia cell-derived inflammatory mediators affect the brain–blood barrier (BBB) integrity. The disruption of the BBB, a common feature of neurodegenerative disease [22], leads to the infiltration of immune and plasma cells in the brain parenchyma. Indeed, Th17 cells, via IL-17 secretion, permeabilize the BBB both to soluble molecules and circulating immune cells [23]. Infiltrated immune cells in the CNS release inflammatory mediators which activate microglia and astrocytes and induce the differentiation of suppressive cells [7,24].

### 3.2. MDSCs in Alzheimer’s Disease

Alzheimer’s disease is the most common neurodegenerative disease. Although ageing is the most important risk factor, its aetiology remains unknown. The disease is characterized by neuroinflammation, amyloid-β (Aβ) deposition, neuronal loss, and neurofibrillary tangle formation, leading to cognitive impairment and memory loss. Although the clinical manifestation of AD is a chronic inflammatory disease that mainly affects the brain, by now it is considered a systemic disease, where the immune system plays a crucial role via the neuro-immune axis [25,26]. Stages of AD can be classified according to the Clinical Dementia Rating (CDR), based on a scale of 0–3, equivalent to no dementia (CDR0), questionable dementia (CDR0.5), mild cognitive impairment (CDR1), moderate cognitive impairment (CDR2), and severe cognitive impairment (CDR3). Animal models for the study of AD are widely used; the most common are the APP/PS1 model and the 5xFAD model. APP/PS1 consists of double-transgenic mice expressing a chimeric mouse/human amyloid precursor protein and a mutant human presenilin 1, both directed to CNS neurons. Both mutations are associated with early-onset AD. 5xFAD consists of transgenic mice overexpressing mutant human amyloid-β precursor protein 695 (APP) with the Swedish (K670N, M671L), Florida (I716V), and London (V717I) Familial AD (FAD) mutation along with human presenilin 1 with two FAD mutations (M146L and L286V). 5xFAD mice may be a useful model of intraneuronal amyloid- β 42 induced neurodegeneration and amyloid plaque formation.

Focusing on the immune system, MDSC expansion in AD has been reported. Although they possess powerful suppressive activity, very few data on the role of MDSCs in AD exist, and of those existing, some are controversial. The search method employed in this review produced six results. One consists of bioinformatic research to highlight peculiar gene expression in AD cases [27]. Two papers investigated MDSC expansion in AD patients and looked at their influence on T cells [28,29]. Three works were conducted using murine models. Of these, two studies used the APP/PS1 mouse model, and one used the 5xFAD mouse model. Of those using the APP/PS1 mouse model, one paper investigated the PD-L1/PD1 pathway involvement in MDSC expansion [30], and the other one investigated the proliferation of each MDSC subset at different AD stages, and their relationship with cytokines expression [31]. The research that used the 5xFAD mouse model investigated the influence of MDSCs on CD4+ T cell proliferation [32]. 

In AD patients, the immune response is enhanced. Indeed, the infiltration of MDSCs, Tregs, and Th17 cells, together with other immune cells, is significantly increased. The study of the peculiar gene expression of these cell populations allowed us to identify five hub mitochondria-related differential expression genes MitoDEGs (*BDH1, TRAP1, OPA1, DLD, SPG7* genes) which represent potential pathological biomarkers. Their mRNA expression levels, except for *SPG7*, decreased in AD and negatively correlated with immune cell infiltration in the brain tissue [27].

Moreover, in humans, MDSCs appear to be expanded in the early stages of the disease and decreased in later stages. PMN-MDSCs are expanded in amnestic mild cognitive impairment (aMCI) with respect to middle AD (mAD) blood samples. The expansion of PMN-MDSCs positively correlates with Treg expansion. The IL-1β level is higher in aMCI, thus correlating with PMN-MDSC proliferation. The M-MDSC subset does not change among aMCI, mAD, and healthy subjects. This reflects the unvaried IL-6 level among the three groups [28]. Despite these results, PMN-MDSC and Treg expansion seems not to correlate with IL-10 expression. Indeed, its results are more expressed in healthy subjects, even though PMN-MDSCs are reported to produce IL-10 to induce Treg differentiation in blood.

The M-MDSC subset appears to expand in the early stages of AD. Indeed, it is observed to increase in CDR0.5 patients and even more in CDR1, but drastically decreased in the later stages of CRD2/3. While *ARG1* results to be clearly expressed during the early stages, the PMN-MDSC subset is not analysed, thus it is not clear whether *ARG1* is expressed by M- or PMN-MDSCs in these early stages of AD. Moreover, it seems that only MDSCs from a very early stage, such as CDR0.5, have immunosuppressive capability resulting in a decrease in IL-6 expression. A possible explanation as to why CDR1 and CDR2/3 MDSCs do not inhibit IL-6 expression is that they release IL-6 on their own, thus the IL-6 levels result is high [29].

MDSC expansion occurring in the early stages of pathology suggests an attempt of the immune system to drive the inflammation toward resolution. Indeed, MDSC occurrence is a response to initial inflammation. Whereas the MDSC reduction in AD late stages implies that a pro-inflammatory phase gains the upper hand with disease progression, and MDSCs are no more effective. On the contrary, in the APP/PS1 mouse model of AD, MDSCs behave oppositely. Indeed, both M-MDSC and PMN-MDSC subsets decrease in the early stage (7 months of age), whereas MDSCs increase in the late stage (11 months of age), with respect to healthy control mice. The reduction in M- and PMN-MDSC numbers in the early stage goes at the same pace as IL-6 expression, which is reported to be decreased in the early AD phases with respect to healthy mice, whereas MDSCs increase in the late stage correlates with enhanced expression of pro-inflammatory cytokines, i.e., TNF-α and IL-6 [31]. Given that M-MDSCs produce IL-6, the expansion of MDSCs in the late stage could rely mainly on the expansion of the M-MDSC subset, which correlates with the increase in IL-6 expression.

Although M-MDSCs are observed to be reduced in 7-month-old APP/PS1 mice with respect to healthy animals, the M-MDSC number is higher in 8-month-old APP/PS1 mice compared to healthy mice. Moreover, M-MDSCs are reduced in AD animals when the onset of another infection, via influenza vaccine administration, together with PD-1 inhibition, occurs [30]. Beyond influenza vaccine administration, the PD1 inhibition blocks the PMN-MDSC induction of Tregs, resulting in lesser TGF-β production, resulting, in turn, in a decreased M-MDSC induction, the solely MDSC subset considered in the research. PMN-MDSCs, here, were not analysed. Moreover, others have reported that in 5xFAD mice, in the case of an increased inflammatory state, due to Porphyromonas Gingivalis (Pg) bacterial injection, the M-MDSC number decreases [32]. The reduction in M-MDSCs results in an exacerbation of neuroinflammation and cognitive impairment. On the contrary, a supplement of exogenous M-MDSCs reduces neuroinflammation, immune imbalance, and cognitive impairment in 5xFAD mice. Moreover, in the presence of the Pg bacteria, not only did M-MDSCs diminish in number, but CD4+ T cell proliferation increased, suggesting an anti-inflammatory function.

In light of these results, it appears that MDSC involvement occurs differently in the pathology progression in humans or murine models of AD. Due to differences between mouse and human immunology [33], the evolution of AD may follow a different path in patients compared to murine models. Thus, caution is required in comparing data obtained from mice with those from humans.

### 3.3. MDSCs in Parkinson’s Disease

Parkinson’s disease is the second-most-common neurodegenerative disease. It is characterized by neuroinflammation, the deposition of misfolded α-synuclein aggregates, and the consequent degeneration of dopaminergic neurons in the substantia nigra, resulting in dopamine reduction in the CNS [34]. Dopaminergic neuronal loss results in motor dysfunction and non-motor symptoms. Aging is the most significant risk factor for developing PD, but also genetics is an important component, in addition to the combination of other different factors. Although the exact aetiology of PD remains unknown, recent evidence has reported neuroinflammation as a crucial factor in PD pathogenesis. Despite PD mainly affecting the nervous system, systemic inflammation plays a critical role in the progression of the disorder, where the immune system is heavily involved. 

In neurodegenerative disease, inflammation leads to infiltration and expansion of MDSCs in the nervous system. MDSC function in PD progression may not be exhaustively studied in the murine model due to the late appearance of the M-MDSC subset in relation to the short duration of PD mouse lesions that may be insufficient to induce the M-MDSC phenotype. According to this, the search strategy employed in this review produced five results concerning the MDSCs in PD. No strategy was conducted using the murine model. Three papers investigated MDSC expansion in PD subjects [35,36,37], one study inquired into the MDSC expansion in PD patients and the MDSC-related gene expression [38], and one study investigated immune cells and MDSC-related gene expression in PD through bioinformatics analysis [39].

It came to light that MDSCs and Th17 cells are significantly expanded in peripheral blood from PD patients with respect to healthy individuals [35,36], suggesting that both cell subsets are associated with neuroinflammation. Furthermore, MDSC and Th17 cell levels in peripheral blood from PD patients are positively correlated [36], suggesting that Th17 and MDSCs are both involved in PD progression. Evidence has shown that IL-6, TGF-β, and IL-1 are significantly increased in cases of PD, which reflects the increase in MDSCs. Moreover, MDSCs produce IL-6 and TGF-β, which drive the Th17 differentiation [35]. Infiltrated Th17 cells increase the release of IL-17, an important inflammatory factor associated with the activation of detrimental inflammatory factors like TNF-α and IL-1 by the microglia, thus inflammatory responses quickly spread throughout the brain [36]. The main function of MDSCs is to suppress the immune response, but the induction of Th17 cell differentiation and expansion could increase the inflammation for a limited time, which promotes PD progression. 

Both papers did not specify which MDSC subset they analysed [35,36]. According to the scientific literature, there are no methods to detect PD at its onset [37], thus the detectable MDSC expansion in PD patients may consist solely of M-MDSC subset proliferation. In this case, it is difficult to take into account the PMN-MDSC subset, which mainly acts during the initial phase, before the onset of symptoms, and thus PD diagnosis. 

A focused investigation on the M-MDSC subset showed that it is increased in PD cases. Moreover, all the M-MDSCs express DAT+/TH+. In myeloid cells, dopaminergic proteins such as the dopamine transporter (DAT) and tyrosine hydroxylase (TH) modulate immune functions and attenuate ongoing inflammation. In PD patients, the CNS dopamine neurons expressing the same markers are decreased [37].

According to the recent findings, the MDSC result significantly increased in the peripheral blood of patients with PD compared with healthy individuals. Moreover, MDSC expansion goes at the same pace as the increased expression of immunosuppression-related genes, such as *ARG1*, *IL-10*, and *COX-2* [39], and *SLC18A2*, *L1CAM*, *S100A12*, and *CXCR4*, which moderately correlates with MDSCs [38], suggesting that immune-related genes are involved in the pathogenesis of PD.

### 3.4. MDSCs in Amyotrophic Lateral Sclerosis

Amyotrophic lateral sclerosis, known as Lou Gehrig’s disease, is a neurological disorder that affects motor neurons. The progressive degeneration and death of motoneurons results in muscle weakness and atrophy, progressive paralysis, respiratory failure, and eventually death within 3–6 years. Only 10% of sufferers survive for a decade or more. Although ageing and some environmental factors are risk factors of ALS, the exact aetiology of the disorder is not known, especially concerning sporadic ALS (sALS). However, recent findings highlight the role of neuroinflammation and the involvement of the immune system in disease progression. The immune system participation implicates a neuroprotective phase in the early stage of ALS, and a neurotoxic phase in the later disease stage. [40,41]. Nevertheless, almost nothing is known about MDSC’s role in ALS. The search strategy employed in this review produced only one result, which investigated both MDSC expansion in sALS patients and the immunosuppressive activity outcome on disease progression in the murine model [42].

It emerged that the circulating MDSC number is significantly higher in sALS patients compared to healthy subjects. Moreover, the MDSC population proved to be heterogeneous, suggesting the presence of both MDSC subsets. Despite this finding, MDSC expansion does not correlate with sALS patients’ age, nor with disease severity [42].

Interestingly, the immunosuppressive activity exerted via the enzyme ARG1 by myeloid cells appeared to worsen the sALS state and progression. The suppressive activity of bone marrow-derived myeloid cells (BMDMs) via ARG1 can be activated by the anti-inflammatory interleukin 4 (IL-4). The administration of IL-4 activated BMDMs in the ALS mSODG93A mice model, results in an earlier appearance of disease signs and shorter life expectancy compared to healthy mice [42]. However, currently, there is no knowledge about each MDSC subset’s contribution to sALS’ different stages.

### 3.5. MDSCs in Multiple Sclerosis 

Multiple sclerosis is the most common immune-mediated disorder affecting the central nervous system. It is an autoimmune condition characterized by neuroinflammation, which drives demyelination in the spinal cord and the brain. This disease causes a wide range of symptoms, including problems in limb motion, balance and coordination impairment, and visual impairments. Although genetics and some environmental factors have been suggested as possible causes of MS, its exact aetiology is not yet known.

Much of our current knowledge concerning the auto-immune pathogenesis of MS comes from EAE, the animal model of MS. 

Four types of MS exist, relapsing–remitting MS (RRMS), characterized by individual relapses (exacerbation); primary progressive MS (PPMS), where symptoms gradually become severe; secondary progressive MS (SPMS), a stage of MS which comes after relapsing–remitting MS for many people; and the relapsing progressive MS (RPMS), where the gradual worsening is interrupted by individual relapses.

In all MS types, the neuroinflammation caused by autoimmunity leads to the formation of myelin plaques, which disrupt the electric signals travelling through the neurons. Moreover, MS lesions in the CNS display an important accumulation of myeloid cells, including MDSCs. Despite the occurrence of MDSC infiltration in the CNS, little is known about their role in MS pathogenesis.

The search method employed in this review produced thirty-three results. Fifteen papers investigated MDSC expansion in the MS and its link with MS stages. Eighteen papers examined the influence of MDSC treatment and the disease progression.

Among the fifteen studies investigating the MDSC expansion in MS, two were conducted in human subjects [43,44], three used both human subjects and a mouse model [45,46,47], eight were performed in a mouse model [15,48,49,50,51,52,53,54], and two in vitro, using MDSCs isolated from MS patients or a mouse model of MS [55,56]. The investigations on MDSC-targeting treatment were carried out in murine models or in vitro cells collected from MS patients or EAE animals.

In RRMS, both M- and PMN-MDSC subsets are increased during the relapse with respect to the remitting phase and healthy subjects [43,44,45,47]. Moreover, the greater the number of M-MDSCs in the relapse phase, the faster the recovery. Indeed, a higher percentage of M-MDSCs during the relapse leads to a faster and full recovery, while a low percentage of M-MDSCs in relapse results in partial recovery, suggesting that M-MDSC expansion during the active phase influences disease progression [47,52]. M-MDSC depletion results in the increase in CD4+T and CD8+T cells and auto-reactive T cells, which exacerbate the demyelination [48]. Nevertheless, independent of the percentage of M-MDSCs in relapse, there are no differences in M-MDSC numbers during the remitting phase between patients subjected to full or partial recovery [47]. Moreover, the T cell suppressive capability in RRMS is increased and may depend on Tregs [43]. Therefore, MDSCs are expanded during the active phase of MS (during the pro-inflammatory phase), and they drive the disease towards the anti-inflammatory phase, in which they are less expanded. 

On the other hand, PMN-MDSCs increase in peripheral blood during the first clinical episode which is suggestive of MS. These cells are also detectable in chronic CNS autoimmunity, i.e., PPMS, and are reduced in MS patients who experienced a recent relapse phase. This finding suggests that an inflammatory disease can trigger the PMN-MDSC expansion, and, consequently, these cells decrease during an anti-inflammatory phase [46].

Similarly, during the very early stage in the EAE mouse model of MS, immune cells release multiple mediators which change the microenvironment, resulting in Treg response and MDSC recruitment. Both Tregs and MDSCs have a role in controlling the overall MS severity [53]. Indeed, significant accumulation of G-MDSCs occurs in peripheral lymphoid organs and particularly in the spleen of EAE mice before disease resolution [45]. Moreover, MDSCs reach the peak in the CNS, in the draining cervical lymph nodes [45,49], and in the lungs [15], at the disease onset, and decrease during the anti-inflammatory phase. Furthermore, G-MDSCs express high levels of PD-L1, required to exert their function of Treg induction in vivo. G-MDSC action results in the inhibition of Th1 and Th17 cells priming, suggesting that they have anti-inflammatory activity [45]. Moreover, the higher the abundance of Ly-6C+ cells, namely M-MDSCs, in the peripheral blood and the spleen at disease onset, the milder the clinical course, highlighting how M-MDSCs at the EAE onset are strongly related to the future clinical course severity. Indeed, a higher abundance of M-MDSCs correlates to smaller areas of demyelination and a lower degree of axonal damage [47,54].

In addition, in MS patients, a negative correlation between the PMN-MDSCs and B cells was observed, suggesting suppressive activity of PMN-MDSCs towards B cells.

In EAE mice, PMN-MDSCs appear increased at disease onset and are unremitting during the recovery phase in the cerebral spinal fluid with respect to blood; the opposite to what happens in patients, where PMN-MDSCs remain detectable in blood in the chronic phase.

The depletion of PMN-MDSCs results in a worsening of clinical symptoms in mice, while PMN-MDSC expansion leads to a faster and complete recovery from the disease. In addition, MDSC depletion drives B cell expansion and exacerbation of disease severity [51]. In the same way as it happens in MS patients, PMN-MDSCs negatively correlate with B cells. Therefore, PMN-MDSCs play an important immunosuppressive role towards B cells, avoiding an exacerbation of EAE symptoms [46]. Besides the MDSC regulative role, the chronic neuroinflammation coming with MS may be established due to a weakening of the immunosuppressive power of MDSCs. Indeed, a high level of miR223 expression has been observed in MS patients with respect to healthy subjects. Similarly, miR223 is significantly more expressed in the spinal cord of EAE mice compared to healthy animals. A study on miR-223−/− EAE mice showed that at the peak of the disease, both PMN- and M-MDSCs were significantly higher in the spleen, but only M-MDSCs in the CNS with respect to healthy mice. The increase in MDSCs in miR-223−/− mice matches with the increase in IL-10 and a decrease in IL-17A, suggesting that miR-223 regulates MDSC suppressive functions. This regulation occurs via increasing the expression of *ARG1* and STAT3 [55,56]. 

Considering all the studies conducted on MS, at first glance, contradictory data appear concerning MDSC expansion in PPMS and SPMS. Indeed, both M- and PMN-MDSCs seem to be significantly reduced in SPMS patients compared to RRMS and healthy subjects [43], whereas the distribution of myeloid cells expressing all the typical markers for M-MDSCs is mainly found in high inflammatory areas of both SPMS and PPMS [47]. The deceptive discrepancies may depend on the different tissues analysed. Indeed, the M- and PMN-MDSCs results are reduced in blood samples, thus the circulating cells are reduced, while the M-MDSCs result is accumulated in nervous tissue damage, namely highly inflamed areas.

Beyond the classical immunoregulatory function of MDSCs, these cells seem to determine even the disease resistance in young mice, where a higher frequency of plasmacytoid DCs and myeloid-derived suppressor cells with immunosuppressive properties delay the disease onset [50]. Moreover, also a regenerative role of MDSCs is suggested. Indeed, MDSC density within the plaque of the myelin lesion positively correlates with the density of oligodendrocyte precursor cells (OPCs) in the adjacent periplaque. This indicates a potential role of MDSCs in promoting the mobilization, survival, and proliferation of OPCs, thus promoting myelin formation [54].

#### MDSCs Modulation via Treatments in MS

Treatments to modulate MDSCs have been developed using drugs, natural or endogenous compounds, inhibitors against endogenous molecules, and synthetic glucocorticoids. The efficacy was tested in murine models or in vitro cells collected from MS patients or EAE animals. The effects on MDSC modulation were analysed in different compartments, such as the peripheral blood, the spleen, the draining lymph nodes (dNLs), and the CNS, which consists of the brain and the spinal cord. Since not all the works have investigated the MDSC modulation in all the organs reported or incompatible results were found, it is difficult to understand how MDSC modulation impacts MS progression. The modulations of MDSCs and T cells by treatments are reported in the table below (Table 1). The studies considered are those dealing with at least one MDSC variation in at least peripheral blood, spleen, dNLs, or CNS.

Analysis of treatments applied to MS highlighted that disease worsening occurs when the MDSC number decreases in the spleen, associated with the reduction in CD4+ and CD8+ T cells [58,60].

The attenuation of MS is observed when a reduction in MDSCs occurs in the peripheral blood [17], in particular of the monocytic subset [67,70], whereas the PMN-MDSCs remain unchanged [70] or increase, matching with the enhancement of *ARG1* expression and the corresponding enzyme activity [67]. As regards the spleen, the MS attenuation is accompanied by a general MDSC expansion both in the monocytic and granulocytic subsets [57,59,60,61,63,65,68,71], even though a reduction in MDSCs in the spleen after gemcitabine treatment results in disease amelioration [17]. This correlation between the reduction in MDSCs in the spleen and disease attenuation could be explained by the fact that the drug used reduces the whole number of MDSCs, not specifically in the spleen, thus also, in this organ, this cell type appears reduced. The expansion of MDSCs in the spleen matches the expansion of Tregs and IL-10 expression [59,60,63,72] and the reduction in CD4+ T cells, CD8+ T cells, and Th17 cells [17,57,58,59,60,68,71,72]. MS attenuation is also accompanied by MDSC changes in the draining lymph nodes, where they remain unvaried [57,61] or increase in number in both subsets [63,72] or only concern the PMN- subset [71]. Contrasting results were noticed about MDSC modulation in the CNS during disease attenuation. Some authors have observed an increase in MDSC expansion after treatment [57,63,65,68], especially regarding the PMN-MDSCs [61,62,71], while others have observed a reduction in MDSCs [17,58,60,69,70]. Irrespective of MDSC variation, the amelioration of MS in CNS matches with the expansion of Tregs or IL-10 [59,66] and the reduction in Th17 and IL-17 [17,59,66,68,70,71].

Considering the studies which investigated the role of MDSCs in humans, it resulted that the two MDSC subtypes are characterized by different behaviours in each disease and in different stages. MDSC differences among neurodegenerative diseases are reported in Table 2.

## 4. Discussion

MDSCs represent a heterogeneous myeloid cell population with the potential to regulate immune response. Both the subsets of MDSCs can drive a pro-inflammatory state towards an anti-inflammatory one, suppressing T effector cells, and promoting T regulatory cells. Because this immunosuppressive cell population is highly heterogeneous and plastic, the definition of a precise role in pathogenetic processes is very difficult and challenging. Current studies provide conflicting information about the role of MDSCs in neurodegenerative diseases. Sometimes, these cells can worsen the disease, while other times they can attenuate it. The conflicting results may depend on the biased nature of the sample across the considered studies. Indeed, some selected samples focused on specific patient populations or disease stages, thus limiting the generalizability of findings to broader patient populations or disease contexts, and reducing the external validity of the research. Moreover, studies employed different isolation techniques for assessing MDSCs, or they specifically looked at MDSC changes only in focused organs, overlooking MDSC alteration in the whole organism. This variability can lead to inconsistencies in results and difficulty in comparing findings across studies, reducing the reliability of conclusions drawn from the literature. 

MDSCs appear to expand significantly during the acute phase of neurodegenerative diseases and may drive the inflammation towards resolution [28,35,36,46,47,48,51,52]. These cells decreased during the anti-inflammatory stages. The complexity of diagnosing diseases before symptom onset, in addition to differences in disease progression, makes it difficult to generate a timeline of MDSC changes common to all neurodegenerative diseases. Both in AD and PD, PMN-MDSCs seem to act at the initial phase, driving these disorders toward an anti-inflammatory state, and to be then replaced by M-MDSCs in the middle/advanced stages of the disease [28,35,36]. In the late stage of PD, M-MDSC expansion may increase the inflammation for a limited time and support the disease progression. In the very advanced phases of the diseases, it seems that MDSCs are no more involved in the disease progression [29]. On the contrary, in ALS, MDSC expansion and immunosuppression seem to exacerbate the symptoms of the disease [42]. However, further investigation is necessary to confirm MDSC’s role in ALS progression. In MS, the initial tremendous accumulation of MDSCs in the CNS to contrast the inflammation and the autoimmunity, appears crucial for a fast recovery from a relapse [46,47,48,51,52]. The consequent accumulation of MDSCs in peripheral organs, accompanied by their reduction in the CNS, seems fundamental to attenuate the symptoms [15,45,47,49,54]. Eventually, the chronic neuroinflammation coming with MS may be related to the weakening of the immunosuppressive power of MDSCs [55,56]. Relationships among neuroinflammation, MDSCs, and disease progression are recapitulated in Figure 4. Beyond the classical immunoregulatory function of the MDSCs, these cells may be capable of supporting myelin regeneration, promoting the mobilization, survival, and proliferation of OPCs in the periplaques [54]. 

Animal models have brought great assistance in understanding the immunosuppressive capability of MDSCs and their involvement in neurodegenerative diseases. Nonetheless, contrasting results were observed about MDSC expansion and the relative disease outcomes, in murine models or humans affected by neurodegenerative diseases. Due to differences between mouse and human immunology [33], MDSC involvement may occur differently in the disease progression in humans or murine models. Indeed, genomic differences between humans and rodents may differently influence neurodegenerative diseases. In addition, transgenic murine models may not entirely recapitulate the complexity of human neurodegenerative diseases, omitting some crucial aspects. And, no less important, even the short lifespan of rodents contributes to the incomplete development of all the pathological aspects in ageing-related neurodegenerative diseases. Therefore, murine models are certainly crucial and fundamental for studying immune cells’ involvement in neurodegenerative diseases, but human and rodent differences should be considered to analyse the obtained results. All these discovered limitations make it difficult to establish causality between MDSCs and neurodegenerative diseases. Undoubtedly, much more research is necessary to elucidate how each subset of MDSCs is involved in neurodegenerative diseases and to highlight how their immunosuppressive activity influences disease progression. To elucidate the MDSC participation and influence on disease progression, future works should consider the two subtypes of MDSC and their relative expansion in CNS and peripheral organs during different stages of the disease.

## 5. Conclusions

The involvement of the immune system in neurodegenerative diseases leads to MDSC proliferation, which influences disease progression. The initial MDSC proliferation acts to resolve the inflammatory state. However, in chronic neuroinflammation, MDSC immunosuppressive capability abates. This condition could lead to an exacerbation of the inflammatory state and eventually participate in the impairment of cognitive processes. The two subtypes of MDSCs behave differently during the AD, PD, ALS, and MS progression, but we do not have complete knowledge about it. Therefore, future studies should bridge a fundamental gap to highlight the exact MDSCs’ influence on neurodegenerative disease progression.

## Figures and Tables

**Figure 1 cells-13-00643-f001:**
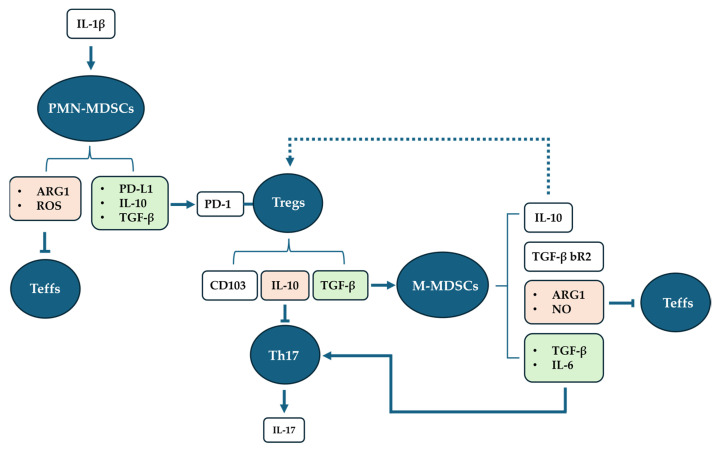
The figure recapitulates the MDSC differentiation process and function. In the presence of IL-β, the PMN-MDSC subset differentiates, and starts to inhibit Teffs via ARG1 expression and ROS production. Simultaneously, PMN-MDSCs express IL-10 and TGF-β, whose presence is crucial for PMN-MDSCs to allow Treg induction and expansion via the PD-L1/PD1 pathway. Once Tregs are activated by the bond between PD1 and its ligand PD-L1, Tregs express CD103, a hallmark of cellular activation, IL-10 through which the inhibition of Th17 is exerted, and TFG-β, necessary for M-MDSC differentiation, is induced by Tregs. M-MDSCs respond to the differentiation increasing the expression of TGF-β bR2. Once differentiated, they exert immunosuppressive activity on Teffs via ARG1 and NO. IL-10 expression could sustain the Treg expansion, acting as positive feedback. M-MDSCs induce Th17 cell expansion via the expression of TGF-β and IL-6. Differentiated Th17 cells produce pro-inflammatory IL-17. The dotted line indicates the hypothesis. The solid lines indicate established pathways.

**Figure 2 cells-13-00643-f002:**
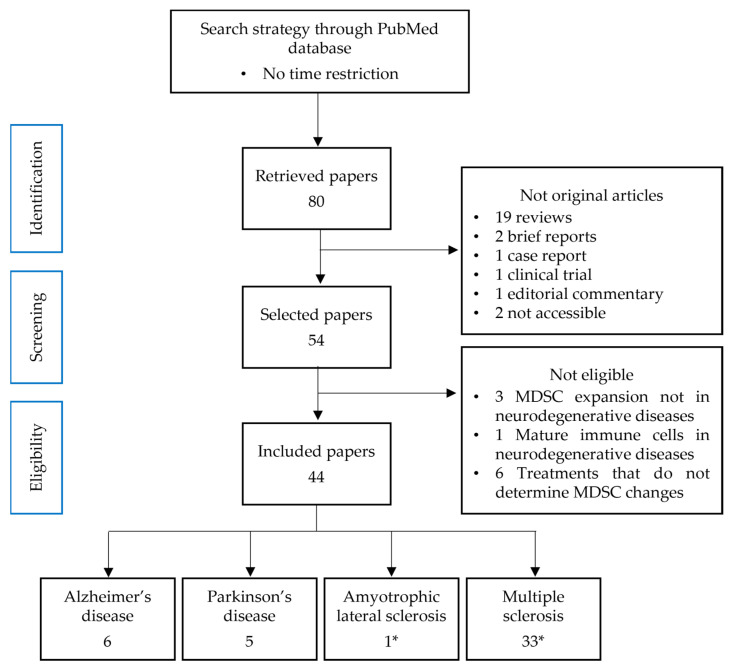
Flowchart of article selection. * One paper discusses both ALS and MS.

**Figure 3 cells-13-00643-f003:**
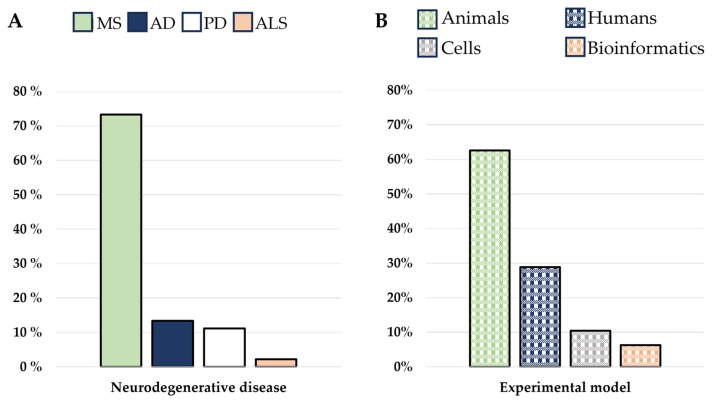
The figure shows the distribution of results found with the research method employed. (**A**) The distribution of publications debating each neurodegenerative disease. Multiple sclerosis (MS), Alzheimer’s disease (AD), Parkinson’s disease (PD), and amyotrophic lateral sclerosis (ALS). (**B**) The distribution of experimental models used in studying MDSCs in neurodegenerative diseases, namely, animals, humans, cells, and bioinformatics.

**Figure 4 cells-13-00643-f004:**
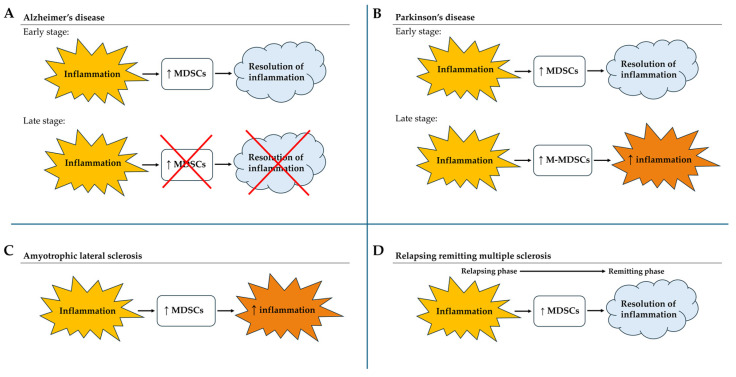
The figure recapitulates the relationships among the neuroinflammation, the MDSCs, and the disease progression. (**A**) In Alzheimer’s disease, the inflammation in the early phase of the disease induces an increase in MDSCs, resulting in the resolution of the inflammation. In the late stage of the disease, the inflammation is no longer associated with MDSC proliferation, and the inflammation persists. (**B**) In Parkinson’s disease, the inflammation in the early phase of the disease induces an increase in MDSCs, resulting in the resolution of the inflammation. In the late stage, the inflammation induces the proliferation of the M-MDSC subset, resulting in the worsening of inflammation. (**C**) In amyotrophic lateral sclerosis, inflammation induces the proliferation of MDSCs, resulting in the worsening of the inflammatory state. (**D**) In relapsing–remitting multiple sclerosis, during the relapse, the inflammation induces the proliferation of MDSCs, which seems fundamental for moving towards the remitting phase.

**Table 1 cells-13-00643-t001:** The table recapitulates MDSCs and T cells’ main changes in four different compartments, namely the peripheral blood, the spleen, the dNLs, and the CNS. The outcomes of the treatment on the myelination state and the MS severity are reported as well.

Treatment	Type of Treatment	Peripheral Blood	Spleen	dNLs	CNS	Myelination State	MS Severity	Ref.
a-GalCer	Immunostimulant glycolipid derived from marine sponge.	-	↑ M-MDSCs ↓ CD4+ T	= MDSCs = CD4+ T	↑ MDSCs ↓ CD4+ T	-	Amelioration	[57]
Am80	Differentiation agent	-	↓ MDSCs ↑ CD4+ T ↓ CD8+ T	-	↓ MDSCs ↑ CD4+ T ↓ CD8+ T	-	Worsening	[58]
anti-PC	Ab against anticoagulant Protein C	-	↑ MDSCs ↑ Tregs ↓ CD4+ T	-	↑ Tregs ↑ IL-10 ↓ Il-17	Improved	Amelioration	[59]
CBD	Non-psychoactive cannabinoid	-	↑ MDSCs ↑ IL-10 ↓ IL-17	-	↓ MDSCs -	-	Amelioration	[60]
CBD	Non-psychoactive cannabinoid	-	↑ M-MDSCs = PMN-MDSCs	= MDSCs	↑ PMN-MDSCs = M-MDSCs	-	Amelioration	[61]
Ch25h	Ablation of oxidoreductase enzyme	-	-	-	↑ PMN-MDSCs	-	Amelioration	[62]
EDI with MBPAc1-9(4Y)	Myelin basic protein MBP_Ac1-9_(4Y)	-	↑ PMN-MDSCs ↑ Tregs	↑ MDSCs	↑ MDSCs	-	Amelioration	[63]
Fingolimod	Immuno- suppressive drug	Treatment results in a greater MS amelioration in patients having higher M-MDSCs amount at the start of the treatment.				Improved	Amelioration	[64]
Gemcitabine	Antineoplastic drug	↓ MDSCs	↓ MDSCs ↓ Th17	↓ Th17	↓ MDSCs ↓ Th17	Improved	Amelioration	[17]
IFN-β	Interferon	-	↑ MDSCs	-	↑ MDSCs	-	Amelioration	[65]
MDSC-PGE2	MDSCs differentiated with prostaglandin (PG)E2	-	= Th17	-	↓ Th17/IL-17 ↑ Tregs	-	Amelioration	[66]
MPPT (humans)	Methylpredniso-lone, a synthetic glucocorticoid	↑ PMN-MDSCs ↓ M-MDSCs ↑ ARG1	-	-	-	-	Amelioration	[67]
NAD+	Nicotinamide adenine dinucleotide	-	↑ MDSCs ↑ ARG1 ↓ Th17/IL-17	-	↑ MDSCs ↑ ARG1 ↓ Th17/IL-17	-	Amelioration	[68]
OM-MOG	Oxidized form of fungal mannan polysaccharide conjugated to myelin antigen	= MDSCs	= MDSCs	-	↓ MDSCs	Improved	Amelioration	[69]
PLY	Antiviral agent with antileukemic activity	↓ M-MDSCs = PMN-MDSCs	-	-	↓ MDSCs ↓ Th17/IL-17	-	Amelioration	[70]
RB6-8C5	Anti Gr-1 Ab	-	↓ MDSCs	-	-	-	Worsening	[60]
SA-IL-4	Serum albumin fused to IL-4	-	↑ PMN-MDSCs ↓ Th17/IL-17	↑ PMN-MDSCs ↓ M-MDSCs ↓ Th17	↑ PMN-MDSCs ↓ Th17	Improved	Amelioration	[71]
SNJ-1945	Calpain inhibitor	-	↑ Tregs ↓ Th17 ↓ IL-17	↑ MDSCs ↑ Tregs ↑ IL-10 ↓ Th17/IL-17	-	Improved	Amelioration	[72]

**Table 2 cells-13-00643-t002:** MDSC changes and function in humans, in different neurodegenerative diseases, such as Alzheimer’s disease, Parkinson’s disease, Amyotrophic lateral sclerosis, and multiple sclerosis.

Disease		Stage of Disease	Cells	Function
Alzheimer’s disease		aMCI	↑ PMN-MDSCs = M-MDSCs	Anti-inflammatory in the early phase. Not involved in later stages
		CDR0.5	↑M-MDSCs	
		mAD	= M-MDSCs	
		CDR1	↑ M-MDSCs	
		CDR2/3	↓ M-MDSCs	
Parkinson’s diseases		Late stages	↑ M-MDSCs	May increase the inflammation for a limited time
Amyotrophic lateral sclerosis		Any stage	↑ MDSCs	Worsen disease progression
Multiple sclerosis	RRMS	Relapse	↑ PMN-MDSCs ↑ M-MDSCs	The greater the number of M-MDSCs in the relapse phase, the faster the recovery
		Remitting	↓ PMN-MDSCs ↓ M-MDSCs	Establishment of an anti-inflammatory state
	PPMS (chronic)		MDSCs detectable	M-MDSCs are associated with chronic inflammation and show reduced suppressive activity
	SPMS (chronic)		MDSCs detectable	M-MDSCs are associated with chronic inflammation and show reduced suppressive activity

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
