# Peer review of "The Influence of Myeloid-Derived Suppressor Cell Expansion in Neuroinflammation and Neurodegenerative Diseases"

_cells, 2024, doi:10.3390/cells13070643_

Round 1

Reviewer 1 Report

Comments and Suggestions for Authors

This review provides a comprehensive overview of the current literature around MDSCs in neuroinflammation and neurodegeneration. The structure and presentation of information allows the main themes to be easily followed.
The Introduction establishes the context around MDSCs and their role in neuroinflammation and neurodegenerative diseases. It provides a comprehensive overview of MDSC phenotypes and the dual inflammatory and immunosuppressive functions of the different subsets. It discusses the potential mechanisms through which MDSCs may influence disease progression.
The Materials and Methods section appropriately describes the search strategy, data collection process, and selection criteria to identify relevant literature.
The Results section is logically organized. It first discusses the participation of the immune system in neurodegeneration to provide context. Then, it analyzes the findings related to MDSCs separately for each major neurodegenerative disease. Within each disease, relevant human and animal studies are summarized and compared in a clear manner.

Minor concerns
The Discussion section could be improved by providing a more comprehensive discussion of the limitations inherent in the reviewed studies. Some of these limitations include the variability in methodologies across studies, potential biases in sample selection, and the reliance on animal models that may not fully recapitulate human disease pathophysiology. Additionally, the discussion could explore limitations related to the interpretation of results, such as the difficulty in establishing causality or the potential confounding factors that may influence the observed associations between MDSCs and neurological diseases.
Here there are some examples:
Studies might employ different methodologies for assessing MDSCs, such as diverse phenotypic markers or isolation techniques. This variability can lead to inconsistencies in results and difficulty in comparing findings across studies, reducing the reliability of conclusions drawn from the literature.
Some studies migh have a bias in sample selection, such as focusing on specific patient populations or disease stages. This could limit the generalizability of findings to broader patient populations or disease contexts, reducing the external validity of the research.

The Authors emphasized the need to consider differences in the immune systems of animals and humans when interpreting the results obtained. However, another limitation of animal models is their potential inability to fully replicate the complexity of human diseases, thereby restricting the translational applicability of findings to clinical settings.

Establishing causality between MDSCs and neurological diseases can be challenging due to the observational nature of many studies. Factors like confounding variables or reverse causality might influence the observed associations, affecting the reliability of conclusions.

Finally, it's worth considering the potential publication bias favoring studies with significant results, potentially distorting the overall interpretation of the literature and impacting the reliability of meta-analyses or systematic reviews.

Author Response

Thanks to the reviewer we implemented the discussion according to the suggestions and we modified the discussion section at lines 528-536 as follows “The conflicting results may depend on the biased nature of the sample across the considered studies. Indeed, some selected samples focused on specific patient populations or disease stages, thus limiting the generalizability of findings to broader patient populations or disease contexts, reducing the external validity of the research. Moreover, studies employed different isolation techniques for assessing MDSCs, or they specifically looked at MDSC changes only in focused organs, overlooking MDSC alteration in the whole organism. This variability can lead to inconsistencies in results and difficulty in comparing findings across studies, reducing the reliability of conclusions drawn from the literature.”

At lines 577-582 as follows “Indeed, genomic differences between humans and rodents may differently influence neurodegenerative diseases. In addition, transgenic murine models may not entirely recapitulate the complexity of human neurodegenerative diseases, omitting some crucial aspects. And, no less important, even the short lifespan of rodents contributes to the incomplete development of all the pathological aspects in ageing-related neurodegenerative diseases.”

At lines 584-586 as follows “human and rodent differences should be considered to analyse the obtained results. All the found limitations make it difficult to establish causality between MDSCs and neurodegenerative diseases”.

Reviewer 2 Report

Comments and Suggestions for Authors

This is a review of the possible role of myeloid-derived suppressor cells (MDSCs) in neurodegenerative diseases - Alzheimer's, Parkinson's, ALS and Multiple sclerosis. The authors have reviewed relevant papers to draw conclusions on the impact of MDSCs in the above-mentioned neurodegenerative disorders. There are some concerns that need to be addressed.

1. in line 255 the authors refer to MDSC occurrence as an "anti-inflammatory phase". This is not completely correct for these diseases. MDSC occurrence is a response to inflammation and not necessarily an anti-inflammatory phase. The authors should make this clear in the text.

2. Line 276, the authors refer to P. gingivalis as a virus. P. gingivalis is a bacteria nota  virus.

3. One or maybe 2 figures are needed to summarize the important points brought up in the Discussion.

4. The authors need to look critically at the interventions that are in Table 1. Specifically, the authors need to discuss the differences in outcome when the intervention specifically targets MDSCs. versus general interventions that target the immune system. Essentially when the immune system is generally targeted, the MDSCs tend to be reduced as a result of beneficial changes in the inflammatory state and the disease is ameliorated. When you use GR1 antibody (fore example), this targets the MDSCs directly, and the outcome appears to be worse. This needs to be discussed. 

5. Figure 1 legend; solid lines indicate "certainty". This needs to be changed to "indicates established pathways". 

6. The authors have numerous grammatical errors throughout the manuscript. It is strongly suggested that the authors sue the services of an English speaking [person(s) to correct the manuscript. 

Comments on the Quality of English Language

needs major English Language correction, throughout the manuscript

Author Response

This is a review of the possible role of myeloid-derived suppressor cells (MDSCs) in neurodegenerative diseases - Alzheimer's, Parkinson's, ALS and Multiple sclerosis. The authors have reviewed relevant papers to draw conclusions on the impact of MDSCs in the above-mentioned neurodegenerative disorders. There are some concerns that need to be addressed.

  1. in line 255 the authors refer to MDSC occurrence as an "anti-inflammatory phase". This is not completely correct for these diseases. MDSC occurrence is a response to inflammation and not necessarily an anti-inflammatory phase. The authors should make this clear in the text.

REPLY: We agree with the reviewer, and we proceeded with an amelioration of the sentence at lines 255-259 as follows: “MDSC expansion occurring in the early stages of pathology suggests an attempt of the immune system to drive the inflammation toward resolution. Indeed, MDSC occurrence is a response to initial inflammation. Whereas the MDSC reduction in AD late stages implies that a pro-inflammatory phase gains the upper hand with the disease progression, and MDSCs are no more effective”.

  1. Line 276, the authors refer to P. gingivalis as a virus. P. gingivalis is a bacteria not a virus.

REPLY: We corrected this error.

  1. One or maybe 2 figures are needed to summarize the important points brought up in the Discussion.

REPLY: Thanks to the reviewer we created an additional figure (figure 4) to summarize and clarify the discussion section.

Figure 4. The figure recapitulates the relationships among the neuroinflammation, the MDSCs and the disease progression. A) In Alzheimer's disease, the inflammation in the early phase of the disease induces an increase of MDSCs, resulting in the resolution of the inflammation. In the late stage of the disease, the inflammation is no longer associated with MDSC proliferation, and the inflammation persists. B) In Parkinson's disease, the inflammation in the early phase of the disease induces an increase of MDSCs, resulting in the resolution of the inflammation. In the late stage, the inflammation induces the proliferation of the M-MDSC subset, resulting in the worsening of inflammation. C) In amyotrophic lateral sclerosis, inflammation induces the proliferation of MDSCs, resulting in the worsening of the inflammatory state. D) In relapsing-remitting multiple sclerosis, during the relapse the inflammation induces the proliferation of MDSCs, which seems fundamental for moving towards the remitting phase.

And we added “Relationships among neuroinflammation, MDSCs and disease progression are recapitulated in Figure 4” at lines 555-556.

  1. The authors need to look critically at the interventions that are in Table 1. Specifically, the authors need to discuss the differences in outcome when the intervention specifically targets MDSCs. versus general interventions that target the immune system. Essentially when the immune system is generally targeted, the MDSCs tend to be reduced as a result of beneficial changes in the inflammatory state and the disease is ameliorated. When you use GR1 antibody (for example), this targets the MDSCs directly, and the outcome appears to be worse. This needs to be discussed.

REPLY: According to the reviewer, we added the following sentence “The studies considered are those dealing with at least one MDSC variation in at least peripheral blood, spleen, dNLs or CNS.” at lines 484-485, to better elucidate this issue.

However, the aim of Table 1 is to recapitulate the disease outcome and the corresponding MDSC variation, after treatments, without establishing any causality between the events. Analyzing the works debating MDSC variations occurring in the condition of disease amelioration, it appears that it is not always true that “MDSCs tend to be reduced as a result of beneficial changes in the inflammatory state and disease amelioration”. Indeed, MDSCs tend to be increased in the spleen as reported at lines 495-496 “As regards the spleen, the MS attenuation is accompanied by a general MDSC expansion both in the monocytic and granulocytic subsets [57,59–61,63,65,68,71]”

 Whereas MDSCs sometimes tend to be reduced and sometimes tend to be increased in the CNS as reported at lines 505-511 “Contrasting results were noticed about MDSC modulation in the CNS during disease attenuation. Some authors have observed an increase in MDSC expansion after the treatment [57,63,65,68], especially regarding the PMN-MDSCs [61,62,71] while others have observed a reduction of MDSCs [17,58,60,69,70]. Irrespective of MDSC variation, the amelioration of MS, in CNS matches with the expansion of Tregs or IL-10 [59,66] and the reduction of Th17 and IL-17 [17,59,66,68,70,71].”

Nevertheless, the disease amelioration is always associated with the expansion of Tregs and IL-10.

  1. Figure 1 legend; solid lines indicate "certainty". This needs to be changed to "indicates established pathways".

REPLY: Thanks to the reviewer we modified the caption of Figure 1 according to his/her suggestion.

  1. The authors have numerous grammatical errors throughout the manuscript. It is strongly suggested that the authors sue the services of an English speaking [person(s) to correct the manuscript.

REPLY: According to the reviewer, we improved the English language and corrected all the grammatical errors.